# Soluble Epoxide Hydrolase Contributes to Cell Senescence and ER Stress in Aging Mice Colon

**DOI:** 10.3390/ijms24054570

**Published:** 2023-02-26

**Authors:** Weicang Wang, Karen M. Wagner, Yuxin Wang, Nalin Singh, Jun Yang, Qiyi He, Christophe Morisseau, Bruce D. Hammock

**Affiliations:** Department of Entomology and Nematology, and UC Davis Comprehensive Cancer Center, University of California, One Shields Avenue, Davis, CA 95616, USA

**Keywords:** colon aging, soluble epoxide hydrolase, cell senescence, ER stress

## Abstract

Aging, which is characterized by enhanced cell senescence and functional decline of tissues, is a major risk factor for many chronic diseases. Accumulating evidence shows that age-related dysfunction in the colon leads to disorders in multiple organs and systemic inflammation. However, the detailed pathological mechanisms and endogenous regulators underlying colon aging are still largely unknown. Here, we report that the expression and activity of the soluble epoxide hydrolase (sEH) enzyme are increased in the colon of aged mice. Importantly, genetic knockout of sEH attenuated the age-related upregulation of senescent markers *p21*, *p16*, *Tp53*, and *β-galactosidase* in the colon. Moreover, sEH deficiency alleviated aging-associated endoplasmic reticulum (ER) stress in the colon by reducing both the upstream regulators *Perk* and *Ire1* as well as the downstream pro-apoptotic effectors *Chop* and *Gadd34*. Furthermore, treatment with sEH-derived linoleic acid metabolites, dihydroxy-octadecenoic acids (DiHOMEs), decreased cell viability and increased ER stress in human colon CCD-18Co cells in vitro. Together, these results support that the sEH is a key regulator of the aging colon, which highlights its potential application as a therapeutic target for reducing or treating age-related diseases in the colon.

## 1. Introduction

Maintaining the normal function and homeostasis of the gastrointestinal (GI) tract is critical for the health and survival of organisms. However, the progressive decline of GI tract function with aging not only leads to local disorders and causes dyspepsia, irritable bowel syndrome, or even colon cancer, but this decline also contributes to the dysfunction in many other organs, including the brain, liver, and bone [1]. Indeed, the chronic inflammation or α-synuclein accumulation in the colon contributes to the pathogenesis of Parkinson’s disease [2]. Additionally, colonic leakage and dysbiosis play key roles in promoting steatohepatitis, cirrhosis, and hepatocellular carcinoma [3]. Thus, a better understanding of the molecular mechanisms underlying gut aging may lead to new therapeutic strategies for age-related pathologies and extend a healthy lifespan.

Cellular senescence is a hallmark of aging and contributes to age-associated tissue deterioration and organ dysfunction [4]. Featured by irreversible cell-cycle arrest and increased apoptotic response, cellular senescence is initiated and regulated by sustained activation of the p53/p21 and p16 pathways [5]. p21 acts as the primary inducer of cell-cycle arrest by mainly inhibiting cyclin E-cyclin dependent kinase (CDK) 2 complex and proliferating cell nuclear antigen [6]. p21 is translationally regulated by p53 and the response to upstream senescence signals, such as intrinsic/extrinsic stress or damage. Similar to p21, p16 also functions as a cell-cycle modulator that targets the cyclin D-CDK4/6 complexes [4]. Cessation of cell division caused by activation of the p53/p21 or p16 pathway will further lead to chromatin budding, histone proteolysis, apoptosis, and senescence-associated secretory phenotype (SASP) release in the senescence cells [7]. In particular, both p53/p21 and p16 pathways are associated with the age-related decline or pathologies in the colon of aged human or rodent models [8,9]. Thus, targeting senescent regulators, especially p21, p16, or/and p53, is a promising strategy to prevent the formation of senescent cells and alleviate age-related tissue deterioration.

In addition to cellular senescence, aging is associated with the functional decline of endoplasmic reticulum (ER) chaperones and folding enzymes, leading to the impairment of proteostasis [10]. The accumulation of misfolded proteins will lead to unresolved ER stress, which is featured by prolonged activation of upstream regulators, including protein kinase RNA (PKR)-like ER kinase (PERK), inositol-requiring enzyme 1 (IRE1), and activating transcription factor 6 (ATF6). These upstream regulators will further trigger the inflammation response by activating the NF-κB pathway or give rise to cell apoptosis via upregulating of pro-apoptotic effector C/EBP homologous protein (CHOP) and growth arrest and DNA damage-inducible gene 34 (GADD34) [11]. Indeed, the levels of CHOP and GADD34 are induced with age in multiple organs [12,13,14]. The induction of CHOP or GADD34 will further sensitize the aged cells to ER- or oxidative stress-induced cell injury and apoptosis [14,15]. On the contrary, suppressing the function of PERK in intestinal stem cells restores gut homeostasis and extends the lifespan of aged *Drosophila melanogaster* [16]. Moreover, ER stress has been found to regulate senescent marker p21 and promote premature aging [17]. Together, ER stress acts as a dominant player in accelerating the aging process.

Soluble epoxide hydrolase (sEH; ephx2), a lipid-metabolizing enzyme, participates in the pathogenesis of numerous inflammatory diseases [18,19]. sEH rapidly converts epoxy fatty acids, which are endogenous lipid mediators formed by cytochrome P450s with numerous beneficial effects, including resolution of inflammation and ER stress, into the corresponding often pro-inflammatory diols, especially dihydroxyoctadecenoic acids (DiHOMEs) [20]. Recent studies show that sEH acts as a key regulator of age-related diseases, including cardiovascular disease, osteoarthritis, and neurodegeneration diseases [21,22,23,24,25,26,27]. In particular, sEH deficiency or inhibition has been reported to reduce cardiac hypertrophy, alleviate stroke, speed recovery, and enhance endothelial function in aortic rings in aged rodents [21,22,23]. Treatment with an sEH inhibitor also suppressed osteoarthritis in aged dogs [24]. Moreover, blocking the function of sEH has attenuated the pathology and progression of neurodegenerative diseases, especially Alzheimer’s and Parkinson’s diseases [25,26,27]. However, the extent of sEH involvement in the aging of the colon is still unknown. 

To this end, the expression and activity of sEH were determined in the colon under aging conditions using a natural aging model in C57BL/6 mice. Using sEH genetic knockout mice, the functional role of sEH in colon aging was also investigated. Moreover, the effect of sEH-derived linoleic acid (LA) metabolites DiHOMEs was examined on cell viability and ER stress in human normal colon CCD-18Co cells.

## 2. Results

### 2.1. sEH Is Increased in the Colon of Aged Mice

Increased sEH activity has been reported in multiple age-related diseases [21,22,23,24,25,26,27]. To determine whether sEH is also involved in colon aging, the expression and activity of sEH were measured in the colon of young (3-month-old) and aged (20-month-old) male mice. The quantitative PCR (qPCR) analysis showed the increased expression of *Ephx2* in the colon of aged mice, compared to young mice (Figure 1A). Similar to increased gene expression, the older mice exhibit increased sEH activity in the colon compared to young mice (Figure 1B). Immunohistochemical analysis further supported the increased level of sEH in the aged colon (Figure 1C). In line with the increased level and activity of sEH, there are increased colonic levels of sEH-produced DiHOMEs in aged mice compared to young mice (Appendix A). Together, these results show that both colonic expression and activity of sEH are significantly increased in the aged mice, indicating a potential association with the aging process in the colon.

### 2.2. Genetic Ablation of sEH Attenuates Senescent Markers in the Colon of Aged Mice

Having demonstrated the elevated sEH expression and activity in the colon, its potential functional role in the aging process was then studied in the colon by comparing the expression of cellular senescent markers in young wild-type male mice (3-month-old), aged wild-type male mice (20-month-old), and aged sEH genetic knockout male mice (*Ephx2*^−/−^) mice (20-month-old). The wild-type aged mice exhibit increased expression of multiple senescent markers, including *p21*, *p16*, *Tp53*, and *β-galactosidase*, in the colon compared to young mice (Figure 2A), which is consistent with previous human and animal studies showing the accumulation of the senescence regulators p21, p16, or p53, which are commonly used as biomarkers for colon aging [8,9]. Immunohistochemical analysis confirmed the increased colonic p21 protein level in older mice (Figure 2B). Interestingly, genetic ablation of sEH blocked the upregulation of senescent markers in older mice (Figure 2A,B), supporting a critical direct role of sEH in the age-related senescence program in the colon.

### 2.3. Genetic Ablation of sEH Abolishes Age-Related ER Stress in the Colon of Aged Mice

The sEH has been shown to regulate the progress of numerous diseases via modulating ER stress [28,29,30]. Thus, we tested whether sEH also regulates age-related ER stress in the colon. qPCR analysis showed the increased expressions of both the ER stress upstream regulators *Perk* and *Ire1*, as well as downstream pro-apoptotic effectors *Chop* and *Gadd34* in the colon of aged mice, compared to young mice (Figure 3A,B). Western blotting and immunohistochemical analysis confirmed the increased Chop protein in the colon of old mice (Figure 3C,D). However, these effects were attenuated in older mice with a genetic deletion of sEH (Figure 3A,D). Overall, these results support that sEH is required for the activation of ER stress in the colon during aging.

### 2.4. sEH-Produced DiHOMEs Reduce Cell Viability and Enhance ER Stress In Vitro

To understand the mechanisms by which sEH contributes to the aging process in the colon, the effects of sEH-derived LA diol metabolites, DiHOMEs, were studied in normal colonic CCD-18Co cells. The DiHOMEs reduced the viability of CCD-18Co cells in a dose-dependent manner (Figure 4A), while its precursors, LA and epoxy-octadecenoic acids (EpOMEs), had no significant effect on the cell viability at the same dose (Appendix A). In addition, exposure to DiHOMEs increased the cytotoxicity-associated lactate dehydrogenase (LDH) release from the cell (Figure 4B). Finally, DiHOMEs increased gene expression of ER-stress downstream pro-apoptotic effectors *CHOP* and *GADD34* in CCD-18Co cells (Figure 4C). Together, these results demonstrate that DiHOMEs reduced cell viability and enhanced ER stress in vitro, which could contribute to the pro-aging effect of sEH.

## 3. Discussion

The world’s population is aging at a fast pace. According to WHO, the aged population (>60 years old) will increase from 380 million in 1980 to almost 2.1 billion in 2050, which means more than 1 in 5 people will be 60 years or older by then [31]. Gut aging contributes to many chronic diseases; however, the detailed molecular mechanisms and endogenous regulators are still largely unknown. Here, we showed that age-related activation of cell senescence and ER stress is paralleled by increased expression and activity of sEH in the colon in mice. Moreover, genetic deletion of sEH attenuates the upregulation of cell senescence and ER stress in the colon of aged mice. Furthermore, the administration of DiHOMEs, the most abundant lipid metabolites derived from sEH in colon, decreased cell viability and increased ER stress in vitro. Altogether, the results support that the sEH contributes to the aging process in the colon by modulating the cell-senescence program and ER stress. 

In addition to this study, previous studies reported that sEH plays a key role in the pathology of other aging-related diseases, especially cardiovascular diseases. Indeed, genetic deletion of sEH decreased oxidative stress and blocked aging-related cardiac hypertrophy and function decline [21]. In aged stroke mice, sEH deficiency attenuates cerebral ischemia-caused infarcts [22]. Moreover, the treatment of aortic rings with a sEH inhibitor and 11,12-epoxy-ecosatrienoic acid (EET) decreased the phenylephrine-induced constriction and enhanced the endothelial function in aortic rings from aged rats [23]. In addition to cardiovascular disease, treatment with an sEH inhibitor attenuated osteoarthritis by reducing the pain condition and increasing the activity function in aged dogs [24]. Besides animal models, an increased sEH level is observed in specimens of the left ventricle from aged ischemic cardiomyopathy patients, compared to non-failing control individuals [32]. Together, all these studies highlight a critical role of increased sEH in regulating a series of diseases or disorders under aging conditions, probably by the enzyme’s rapid hydration of inflammation resolving epoxy fatty acids to their less active or even pro-inflammatory diols. A limitation of the current study is that we mainly detected the expression and activity of sEH in colon of young and aged mice, while the conditions of sEH in the mid-aged mice are still unclear. Further studies are needed to better characterize the dynamic changes of expression and activity of sEH in colon at different age stages.

Cell senescence is one of the predominant hallmarks of aging and is controlled by prolonged activation of the p53/p21 or p16 pathways [5]. Here, the aged mice exhibit increased expression of senescent markers, including *p21*, *p16*, *Tp53*, and *β-galactosidase*, in the colon compared to young mice. These observations indicate the enhanced senescence process in the colon of aged mice. These results are consistent with previous studies showing that p21^+^ cells accumulate in the intestine of aging in mice [33]. Moreover, a human study showed that the numbers of p16^+^ and p21^+^ cells increased with age in colon tissue [8]. Elevated levels of p21 and p16 caused cell-growth arrest, which will further cause aging-related organ deterioration by stimulating paracrine senescence, stem-cell dysfunction, and tissue matrix disruption and inflammation [4]. Importantly, in the current study, genetic knockout of sEH decreased the accumulating of senescent markers *p21*, *p16*, *Tp53*, and *β-galactosidase* in the colon from aged mice, indicating the sEH as a novel endogenous regulator for the cell-senescence program under aging. Previous studies support the notion that attenuating senescence markers helps to reduce age-related alterations. Remarkably, suppression of p16 attenuated the stress-induced premature senescence and maintain gut barrier function in mice [34]. Similarly, the removal of p21-expressing cells improves physical function in aged mice [33]. Thus, downregulating senescent markers, especially *p21* and *p16*, by blocking the function of sEH could protect the colon from age-related senescent changes and attenuate the onset of age-related pathologies in colon tissues. Further studies are warranted to better characterize the protein level of senescent markers, especially β-galactosidase and p16, using tissue staining.

ER stress acts as another central molecular mechanism responsible for age-associated pathogenesis. Here, ER stress was found increased in the colon of aged mice. Notably, there are the elevated expressions of both upstream regulators *Perk* and *Ire1* and downstream pro-apoptotic effectors *Chop* and *Gadd34*, indicating the activation of entire ER stress pathways under aging conditions. This finding is consistent with the previous evidence showing the increased levels of Chop and Gadd34 in multiple organs of aging rats [12,13,14]. Moreover, the increased Chop and Gadd34 levels will further promote the apoptosis of aged cells under ER or oxidative stress [14,15]. Promisingly, sEH deficiency suppressed the aging-induced ER stress by downregulating both upstream regulators and downstream pro-apoptotic effectors. This is similar to previous studies showing that sEH contributes to chronic inflammation and tissue damage by modulating ER stress [28,29,30]. Indeed, in the streptozotocin-induced diabetes model, podocyte knockout of sEH diminished hyperglycemia-caused renal dysfunction by inhibiting ER stress in diabetic mice [28]. Moreover, in the CCl_4_-induced liver fibrosis model, blocking sEH prevented hepatic tissue degradation by restraining the activity of all three branches of ER stress response in the liver of mice [29]. Additionally, in a murine periodontitis model, sEH inhibition reduced *Aggregatibacter actinomycetemcomitans* infection-induced inflammatory bone loss by reducing the ER stress response in the gingival tissues of mice [30]. Altogether, this study, along with the previous research, supports that ER stress acts as a key etiology modulated by sEH under numerous disease conditions.

Acting as an eicosanoid metabolizing enzyme, sEH is responsible for converting inflammation-resolving epoxy fatty acids into diols, which in some cases are inflammation-promoting [19]. Here, treatment of sEH-produced DiHOMEs reduced cell viability and increased ER stress in normal colonic cells. In line with this finding, previous studies showed the detrimental role of DiHOMEs in mediating a series of pathological processes in vivo and in vitro. Particularly, treatment of DiHOMEs cause pulmonary epithelial leakage and injury in mice [35]. DiHOMEs also cause immune cell dysfunction [36], increase mitochondrial permeability [37], and promote MCF-7 breast cancer cell proliferation [38]. A recent study showed that DiHOMEs levels are elevated in plasma samples of hospitalized COVID-19 patients [39]. Thus, DiHOMEs could act as lipid metabolites contributing to the effect of sEH in the colon under aging conditions. This observation is alarming considering the massive increase in LA consumption in both the western diet and as a proportion of total fatty acid intake [36,40]. Further studies are needed to investigate the effects of DiHOMEs on age-related colonic dysfunction.

In conclusion, this study demonstrates that the sEH, a key oxylipin-metabolizing enzyme converting inflammation-resolving epoxy fatty acids to their diols, is involved in, and contributes to, the aging process in the colon by enhancing cell senescence and ER stress. These findings highlight an attractive direction to develop sEH inhibition as a novel therapeutic target for aging-related disease intervention, given that a series of sEH inhibitors are under clinical development for pain [41], hypertension [42], chronic obstructive pulmonary disease [43], and osteoarthritis [24]. Further studies are needed to determine the effect of sEH inhibitors on colonic function in aging models. Altogether, the current study identifies the role of the sEH in mediating the aging process in the colon, which could help further studies to establish sEH as a promising pharmacological target in alleviating the age-related dysfunction in the colon under the background of the global aging issue.

## 4. Methods and Materials

### 4.1. Animal Study 

All animal experiments were conducted with animals’ welfare ensured and in accordance with the protocol approved by the Institutional Animal Care and Use Committee (IACUC, protocol # 20863) of the University of California-Davis. The wild-type (WT) or sEH genetic knockout mice (*Ephx2*^−/−^) were maintained in a standard animal facility, as previously described [44]. During the study, all the mice were maintained on a standard chow diet. The young (3-month-old, n = 6 mice per group) and aged (20-month-old) WT (n = 5 mice per group) or sEH knockout male mice (n = 6 mice per group) were euthanized using isoflurane (5%, Dechra Pharmaceuticals, Northwich, UK) inhalation followed by cervical dislocation. The colon tissues were dissected for further analysis.

### 4.2. sEH Activity Measurement

The colon tissues were collected and lysed to measure the sEH activity. sEH activity was measured as previously described [45]. Briefly, [^3^H]-*trans*-diphenyl-propene oxide (t-DPPO) was used as a substrate. To 100 µL of diluted tissue suspension, a 1 µL of a 5 mM solution of t-DPPO in dimethyl sulfoxide (DMSO, Thermo Fisher Scientific, Hampton, NH, USA) was added ([S]_final_ = 50 µM; 10,000 cpm). The mixture was incubated at 37 °C for 30 min, and the reaction was quenched by the addition of 60 µL of methanol (Thermo Fisher Scientific) and 200 µL of isooctane (Thermo Fisher Scientific), which extracts the remaining epoxide from the aqueous phase. The enzyme activity was determined by measuring the quantity of radioactive diol formed in the aqueous phase using a scintillation counter (TriCarb 2810 TR, Perkin Elmer, Shelton, CT, USA). Assays were performed in triplicate. The protein concentration was quantified using the Pierce BCA assay (Pierce, Rockford, IL, USA).

### 4.3. Total RNA Isolation and Quantitative Reverse-Transcriptase DNA Polymerase Chain Reaction (qRT-PCR) Analysis

Total RNA was isolated from the colon tissues or CCD-18Co cells using TRIzol reagent (Ambion, Austin, TX, USA) according to the manufacturer’s instructions. The quality of the extracted RNA was measured using a NanoDrop Spectrophotometer (Thermo Fisher Scientific, Waltham, MA, USA) and was then reverse transcribed into cDNA using a High-Capacity cDNA Reverse Transcription kit (Applied Biosystems, Waltham, MA, USA). qRT-PCR was performed in the Mic qPCR Cycler (Bio Molecular Systems, Upper Coomera, Australia) using Maxima SYBR-green Master Mix (Thermo Fisher Scientific). All the mouse- or human-specific primers were purchased from Thermo Fisher Scientific. The sequences of primers are obtained from the PrimerBank database [46,47,48] with detailed information listed in Appendix A. The results from target genes were normalized to *glyceraldehyde-3-phosphate dehydrogenase* and expressed to the young mice using the 2^−ΔΔCt^ method.

### 4.4. Tissue Staining 

The tissue fixation, paraffin embedding, section, and dewaxing were performed as previously described [49]. Antigen retrieval was performed by heating the sections in 0.01 M citrate buffer (pH 6.0) to 95 °C for 10 min. The immunohistochemistry staining was conducted using horseradish peroxidase (HRP)/3,3′-diaminobenzidine (DAB) Detection IHC kit (Abcam, Cambridge, UK) according to the manufacturer’s instructions. The anti-p21 antibody (Cell Signaling, Danvers, MA, USA, catalog # 64016), anti-Chop antibody (Cell Signaling, catalog # 2895), or anti-sEH antibody (Santa Cruz Biotechnology, Dallas, TX, USA, catalog # sc-166961) was used to probe the target protein in the tissue section. Sections were then counterstained with hematoxylin for 1 min. The staining intensity of p21, Chop, and sEH was analyzed by Image J software using IHC Toolbox.

### 4.5. Protein Extraction and Immunoblotting 

The colon was dissected and homogenized after being frozen by liquid nitrogen. The protein extraction and immunoblotting were performed as described [50]. Briefly, proteins were extracted with radioimmunoprecipitation assay (RIPA) lysis buffer (Boston BioProducts, Milford, MA, USA) containing 50 mM Tris-HCl, 150 mM NaCl, 1% NP-40, 0.5% sodium deoxycholate, 0.1% sodium dodecyl sulfate (SDS), 5 mM ethylenediaminetetraacetic acid, with protease inhibitor cocktail (Boston BioProducts). Protein concentrations were determined using the BCA protein assay kit (Thermo Fisher Scientific). The samples with equal amounts of protein (15 μg) were resolved on SDS-polyacrylamide gel electrophoresis gels (Bio-Rad Laboratories, Hercules, CA, USA) and transferred onto nitrocellulose membrane (Bio-Rad Laboratories). The membrane was blocked in 5% milk buffer for 1 h, incubated with primary antibodies against Chop (Cell Signaling, catalog # 2895) in 5% bovine serum albumin solution at 4 °C overnight. The membranes were incubated with secondary anti-mouse with HRP-linked antibody for 1 h (Cell Signaling, catalog #7076) and then incubated with Clarity Western Enhanced Chemiluminescence (ECL) Substrate (Bio-Rad Laboratories) for imaging. The chemiluminescence was detected using Bio-Rad ChemiDoc™ Imaging Systems and quantification of immunoblotting was performed using Image J software. β-actin (Sigma-Aldrich, St. Louis, MO, USA, catalog # A2228) was used as a loading control. Data are normalized against those of the corresponding β-actin.

### 4.6. Cell Assays

Human normal colonic CCD-18Co cells were obtained from American Type Culture Collection (Manassas, VA, USA). CCD-18Co cells were cultured in Eagle’s Minimum Essential Medium (EMEM, catalog # 10009CV, Corning Inc., Corning, NY, USA) supplemented with 10% fetal bovine serum. All cells were maintained in an atmosphere of 5% CO_2_ and at 37 °C. 

Cell viability assay was conducted as described [51]. In detail, CCD-18Co cells were seeded in 96-well plates at a density of 30,000 cells/well and incubated for 24 h. Cells were then treated with methyl esters of DiHOMEs (1, 10 μM), EpOMEs (10 μM), LA (10 μM), or vehicle control (DMSO, 0.1%) for 24 h. The treatment doses and time were selected based on previous studies that demonstrating the detrimental effect of DiHOMEs in vitro [36,38]. After that, cells were washed with phosphate buffered saline (PBS), 0.5 mg/mL 3-(4,5-dimethylthiazol-2-yl)-2,5-diphenyl-2H-tetrazolium bromide (MTT) (catalog # M6494, Thermo Fisher Scientific) in fresh EMEM was added to each well and then incubated for 2 h at 37 °C. After the supernatant was removed, 100 μL of DMSO was added to each well. The difference in absorbance at 570 nm was measured on a microplate reader (Molecular Devices, San Jose, CA, USA). Results were expressed as percentages of control (%).

For LDH assay, CCD-18Co cells were seeded in 12-well plates at a density of 300,000 cells/well and incubated for 24 h. Cells were then treated with methyl ester of DiHOMEs (10 μM) or vehicle control (DMSO, 0.1%) for 24 h. The cell-culture media from treated CCD-18Co cells were collected and LDH leakage into the media was determined using the LDH Cytotoxicity Detection Kit (catalog # MK401, Takara, Kusatsu, Shiga, Japan) according to the manufacturer’s instructions. Results were expressed as percentages of control (%).

### 4.7. Enzyme-Linked Immunosorbent Assay (ELISA) Analysis

Colon tissue samples were extracted by mixing with methanol and then washed with hexanes as previously stated [52]. Briefly, the extractions were diluted with PBS for the ELISA analysis. The ricinoleic acid (NuChek Prep Inc. Elysian, MN, USA)–ovalbumin (Sigma-Aldrich) antigen (1 μg/mL) was coated on the plate at 4 °C overnight. After blocking with 3% skim milk, 50 μL of the samples (50 times dilution) or DiHOMEs standard solution was equally mixed with 1 μg/mL polyclonal antibody for 1 h at room temperature. After washing, Goat-anti-Rabbit IgG-HRP (Fitzgerald Industries International, Concord, MA, USA) was added for another 1 h. Finally, the 3,3′5,5′-tetramethylbenzidine (Sigma-Aldrich) detection reagent was used for color development and 1 M H_2_SO_4_ (Sigma-Aldrich) was applied to stop the reaction. The absorbance was measured at 450 nm and the final concentrations of DiHOMEs were calculated using the standard curve.

### 4.8. Statistical Analysis

All data are expressed as the mean ± standard error of the mean (SEM). The normality of data was determined by Shapiro–Wilk test and the equal variance of data was checked by Levene’s mean test before the statistical analysis. Statistical comparison of two groups was assessed by either Student’s *t*-test or Wilcoxon–Mann–Whitney test (if the normality test failed). Statistical comparison of three groups was analyzed using one-way ANOVA followed by Tukey’s or Fisher’s post hoc test or using Kruskal–Wallis test on ranks (if the normality test failed) followed by appropriate post hoc test. All data analysis was performed by using SigmaPlot software (Systat Software Inc., San Jose, CA, USA). All the figures were plotted using GraphPad software (GraphPad Software Inc., Boston, MA, USA).

## Figures and Tables

**Figure 1 ijms-24-04570-f001:**
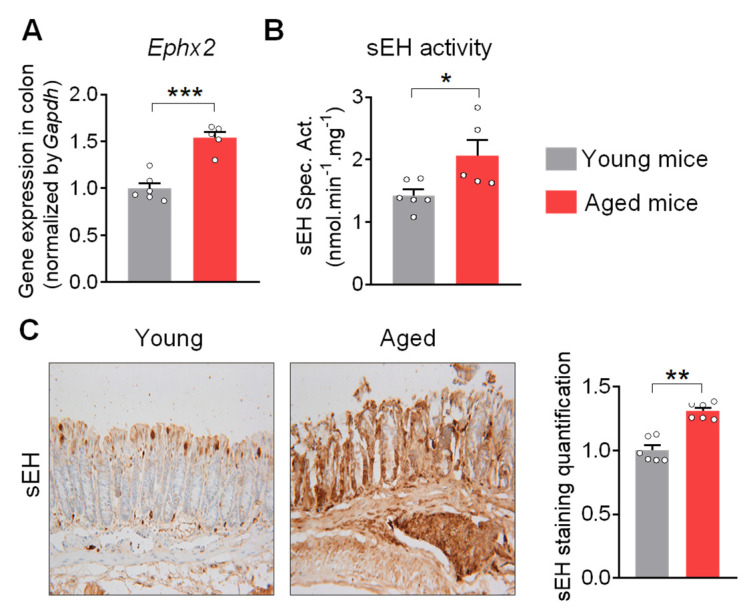
**The expression and activity of sEH are increased in the colon of aged mice.** (**A**) Gene expression of *Ephx2* in the colon. (**B**) Enzymatic activity of sEH in the colon. (**C**) Immunohistochemical staining of sEH (magnification 200×) and quantification of sEH staining intensity in the colon (n = 6 random fields per group) in the colon. The results are expressed as mean ± SEM. n = 5–6 mice per group. Statistical significance of two groups was determined using Student’s *t*-test or Wilcoxon–Mann–Whitney test. * *p* < 0.05, ** *p* < 0.01, *** *p* < 0.001.

**Figure 2 ijms-24-04570-f002:**
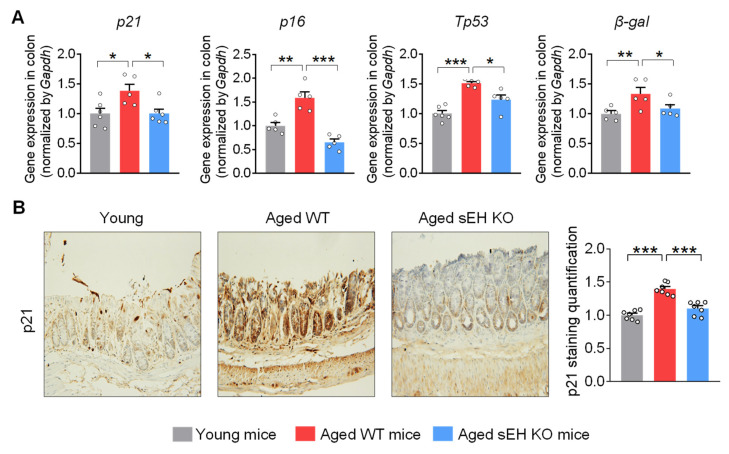
**sEH deficiency attenuates the cell senescent markers in the colon of aged mice.** (**A**) Gene expression of cell senescent markers *p21*, *p16*, *Tp53*, and *β-galactosidase (β-gal)* in the colon. (**B**) Immunohistochemical staining of p21 (magnification 200×) and quantification of p21 staining intensity in the colon (n = 7 random fields per group). The results are expressed as mean ± SEM. n = 5–6 mice per group. Statistical significance was determined using one-way ANOVA or Kruskal–Wallis test on ranks. * *p* < 0.05, ** *p* < 0.01, *** *p* < 0.001.

**Figure 3 ijms-24-04570-f003:**
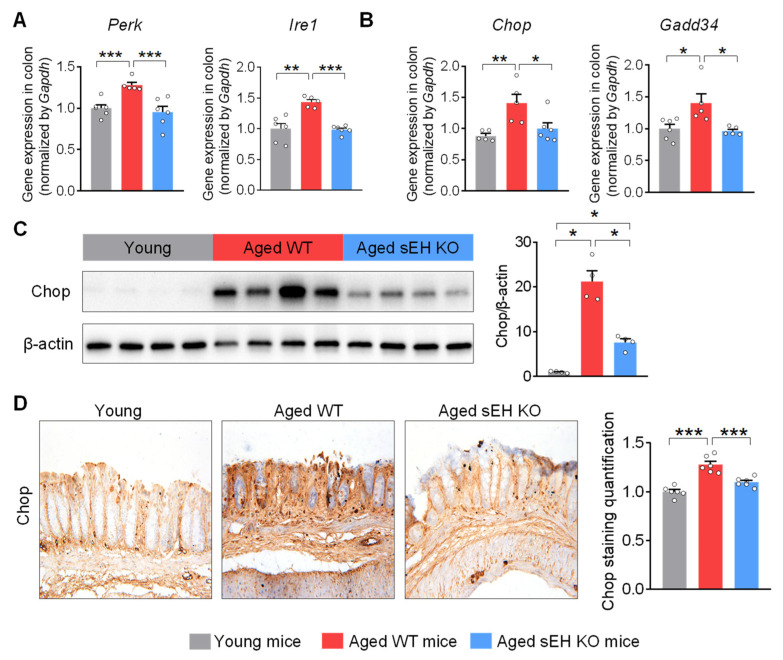
**sEH deficiency alleviates ER stress in the colon of aged mice.** (**A**) Gene expression of ER stress upstream regulator *Perk* and *Ire1* in the colon. (**B**) Gene expression of ER stress downstream pro-apoptotic effector *Chop* and *Gadd34* in the colon. (**C**) Immunoblotting analysis of Chop in the colon (n = 4 mice per group). (**D**) Immunohistochemical staining of Chop (magnification 200×) and quantification of Chop staining intensity in the colon (n = 6 random fields per group). The results are expressed as mean ± SEM. n = 5–6 mice per group. Statistical significance was determined using one-way ANOVA or Kruskal–Wallis test on ranks. * *p* < 0.05, ** *p* < 0.01, *** *p* < 0.001.

**Figure 4 ijms-24-04570-f004:**
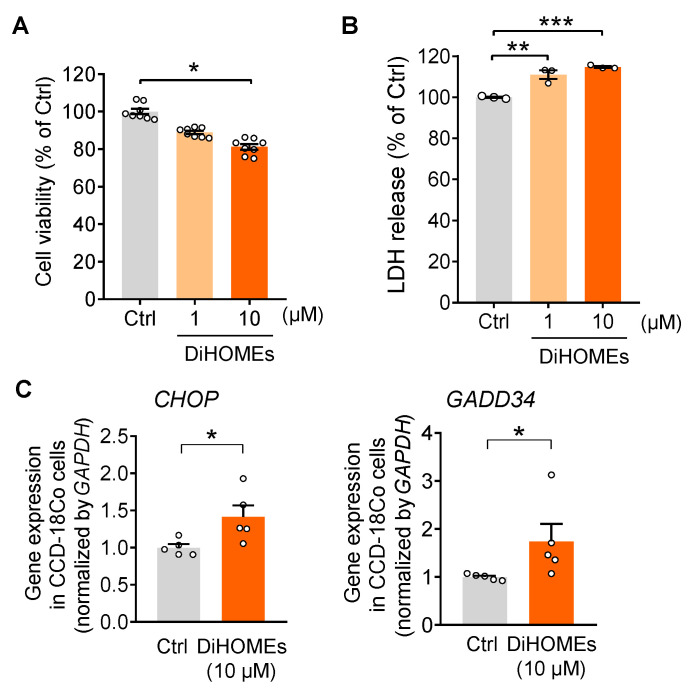
**DiHOMEs reduce cell viability and enhance ER stress in human colonic CCD-18Co cells.** The human colonic CCD-18co cells were treated with DiHOMEs at 1–10 μM for 24 h. (**A**) Cell viability of CCD-18Co cells. (**B**) Lactate dehydrogenase (LDH) release of CCD-18Co cells. (**C**) Gene expression of ER stress downstream pro-apoptotic effectors *CHOP* and *GADD34* in CCD-18Co cells. The results are expressed as mean ± SEM. Statistical significance of two groups was determined using Student’s *t*-test or Wilcoxon–Mann–Whitney test. The statistical significance of three groups was determined using one-way ANOVA. * *p* < 0.05, ** *p* < 0.01, *** *p* < 0.001.

## Data Availability

Please contact the corresponding author to discuss the availability of the data presented in this study.

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
