# Peer review of "Soluble Epoxide Hydrolase Contributes to Cell Senescence and ER Stress in Aging Mice Colon"

_ijms, 2023, doi:10.3390/ijms24054570_

Round 1
Reviewer 1 Report
The aim of this study is to investigate the expression, activity and the functional role of sEH using a natural aging mouse mode. In general, author made a very good attempt to prove the hypothesis. However, there are few questions needs to be addressed.
1. The author investigates the expression and activity of sEH in young and aged mice. It would be good to show the expression of sEH at the mid age?
2. In addition to the β-gal gene expression analysis it would be good to show the β-gal staining in control and experimental tissue samples.
3. Author claimed sEH-produced DiHOMEs reduces cell viability and enhance ER stress in vitro. If possible it would be scientifically important to show the levels in WT and sEH KO mice.
4. The Y axis labels in figures 1A,B, 2B and fig 4 needs to be corrected
5. The statistical significance (*) should be labeled clearly in most of the figures.
6. In Figure 4A, B the font size of the labels in the X axis needs to be increased.
Reviewer 2 Report
M&M:
1. On what basis was the study protocol established? Please assign bibliography.
2. Was a control group used for each study group? This is how the test protocol should be designer (in vivo and in vitro studies).
3. Please assign bibliography to each of the methods used (no self-citation):
- sEH activity measurement
- qRT-PCR
- tissue staining
- protein extraction and immunoblotting and
Cells assays
Were the sequences of the primers used been designer (Table S1)? If not, please provide references. In which company were the primers bought? No information available regarding: product length, characteristic annealing temperatures and references.
4. Animal study:
- Lack of approval from the Ethics Committee.
- Were the animals welfare ensured?
- Experimental protocol: Please present the test groups of animals. How many mice were used for each group. Was a control group used for each study group? This is how the test protocol should be designed.
5. In vitro studies - no experimental model described. On what basis was the study design established? If this is the applicable standard, please refer to the specific literature. On what basis was the dose and time of administration of the substance determined (dose-response, time-course?) - in vitro studies.
6. What program was used to make the figures?
RESEARCH RESULTS – No documentation of gene expression pictures. Should be included in the article or sent to the reviewer.
DISCUSSION – it is recommended to extend it.
CONCLUSIONS – It is recommended to formulate more careful conclusions regarding the clinical aspect of the conducted research.
Reviewer 3 Report
The authors are leaders in sEH inhibitor pharmacology as well as sEH biology. This manuscript reports a role of sEH action on colonic aging; specifically that sEH activity is increased in aged mice and that genetic ablation of sEH attenuates expression of senescent markers in colons of aged mice. A final insight is provided by the report of sEH metabolites have direct toxic effects on CCD-18Co cells that are a normal colon fibroblast cell line. There are only minor changes requested.
In the 5th paragraph of the introduction the second to the last sentence is "EH" meant to be "sEH?"
On several of the bar graphs (Fig 1 A, B, Fig 2B, Fig 4, A, B, C, &D the font on the y axis and numerical values are nearly illegible and should be changed to match those used in Fig 2A, or Fig 3.
In Animal Study description, state how the mice were anethetized and killed. Sacrifice is non-descript and vaguely religious.
Round 2
Reviewer 1 Report
Author addressed the issues adequately
Author Response
We thank the reviewer for all comments and suggestions.